# Survival status and predictors of early neonatal mortality among neonates admitted to neonatal intensive care units in Addis Ababa Public Hospitals, Ethiopia: A retrospective cohort study

Ashenafi Seifu Gesso[1]*, Gemechis Kabe Gonfa[2], Meron Abrar Awol[1]

1 Addis Ababa University College of Health Sciences, Addis Ababa, Ethiopia, 2 Ambo University College of Health Sciences, Ambo, Ethiopia

* seifuashenafi@gmail.com

## Abstract

### Background

The first week after birth is the high-risk time for neonatal death. Early neonatal mortality is still a major global health problem, particularly in sub-Saharan African nations like Ethiopia. Most neonatal deaths (about 75%) occur within the first seven days of life. Still, there is limited evidence on survival status and predictors that may determine when a neonate dies in the early days of life in the study area.

### Objective

The study aimed to assess survival status and predictors of early neonatal mortality for neonates admitted to neonatal intensive care units in the study area, 2023.

### Methods

A retrospective cohort study was conducted in four randomly selected public hospitals from January 1–2022 to January 1–2023 on 375 randomly selected neonatal data charts. The data were collected through document review and checklists using conventional random selection approaches. The collected data was analyzed by SPSS version 26.0. The Kaplan-Meier survival curve with a log-rank test was applied to compare the survival rates between groups. Bivariable and multivariate Cox proportional hazard regression analyses were conducted to identify the predictors of early neonatal mortality. The assumptions of the model were verified.

### Results

A total of 375 eligible neonates were studied and the incidence rate of 37.1 deaths per 1000 live births was found (95% CI: 25.5, 40.9), with 75 of them (20%) died. Premature birth [AHR: 4 (95% CI: 1.234, 5.80)], pre-eclampsia [AHR: 2.4 (95%

**Data availability statement:** The datasets used and/or analyzed during the current study are available from the corresponding author without Data cannot be publicly shared due to ethical restrictions and potentially identifying data. The datasets used and/or analyzed during the current study are available from the corresponding author without restriction. Additionally the data set used for this manuscript can be accessed from the IRB head Mr. Leulayehu Akalu, leulayehu.akalu@aau.edu.et, or from http://aau.edu.et/researchrepository.

**Funding:** The author(s) received no specific funding for this work.

**Competing interests:** The authors have declared that no competing interests exist.

**Abbreviations:** AHR = adjusted hazard ratio, ANC = ante-natal care, CHR = crude hazard ratio, CI = confidence interval, ENM = early neonatal mortality, GA = gestational age, NICU = neonatal intensive care unit, SPSS = statistical package for social sciences.

CI: 1.16, 4.98)], fifth-minute Apgar score [AHR: 3.93 (95% CI: 1.76, 8.77)], not initiating exclusive breastfeeding [AHR: 3.69 (95% CI: 1.14, 9.02)], and low birth weight [AHR: 2.01 (95% CI; 1.28, 3.43) were found to be the independent predictors of early neonatal mortality.

## Conclusion and recommendation

High early neonatal mortality is observed in the study area. Gestational age, pre-eclampsia, 5th min Apgar score, not initiating exclusive breastfeeding, and low birth weight were independent predictors of early neonatal death. We recommend that proactive care for a neonate with a low Apgar score, who is not breastfeeding, is premature, has a low birth weight, and is born from a mother who has pregnancy-induced hypertension is very important.

## 1. Introduction

Neonates are young infants aged after birth up to 28 days of life. They are highly susceptible during this time because significant physiological changes are required for extra-uterine life [1]. Neonatal death is the possibility of a live-born child dying within the first 28 days of life (mortality), regardless of gestational age at birth. Neonatal mortality can also be divided into two: (a) early neonatal deaths, which may happen between the day of birth and the seventh day of life (0–7 days); and (b) late neonatal deaths which happen between the seventh day and the 28th day of life [2]. Neonatal survival during the first week of life is affected not only by the strains of intrauterine existence but also by the birth process, the adaptation to a new environment, nutrition, and infection. Thus, the first seven days of life after birth are the most critical days for life or death [2].

The majority of neonatal deaths occur in low and middle-income nations, where both maternal and neonatal survival depends on the ongoing and improved availability of life-saving therapies. International and local attention must continue to be directed towards the health and survival of every neonate [3]. In sub-Saharan Africa, infections, birth asphyxia, complications of preterm birth, and low birth weight are the most common causes of neonatal mortality. Five countries including Nigeria, the Democratic Republic of Congo, Ethiopia, Tanzania, and Uganda shared 50% of neonatal death in this region [4]. Moreover, in East Africa, this study showed that home birth practices and rural residency are predictors of neonatal mortality.

Ethiopia was ranked 6th among ten countries with the highest neonatal death with 87900 deaths annually and showed slow progress in reducing neonatal mortality rate (NMR). According to the 2019 Ethiopian Mini Demographic and Health Survey (2019 EMDHS) report, Ethiopia's neonatal mortality rate is 30 per 1000 live births [5].

One of the top global healthcare priorities is reducing neonatal mortality, and the Sustainable Development Goals (SDGs) contain specific targets for doing so. As such, the Sustainable Development Goals prioritizes improving mother and child health, to lower global newborn mortality to 12 deaths per 1,000 live births by 2030 [6].

Evidence shows that since 2000, the likelihood of child survival has increased, but not all childhood ages have shown similar progress. Neonates are most likely to die in the first month of life, with an average global rate of 17 deaths per 1,000 live births in 2019, which dropped by 52% from 38 deaths per 1,000 in 1990 [3] and [4].

Worldwide, 2.4 million children died in their first month of life in 2019. Nearly 75% of all neonatal deaths occur in the first week of life, and about a third die within the first day after birth (early neonatal death). The majority (99%) of this occurs in low-income nations. Sub-Saharan Africa accounted for 42% of all newborn deaths while Central and Southern Asia together accounted for 37% of all neonatal deaths [4].

Preterm birth, labor problems, maternal-related conditions, and birth abnormalities are the most frequent causes of neonatal mortality. Infections and complications from premature birth contribute to more than 80% of causes of neonatal mortality [4,7,8].

About 75% of neonatal deaths in low and middle-income countries can be prevented through effective schemes with existing simple and low-cost tools including kangaroo mother care for preterm babies, early breastfeeding initiation, newborn resuscitation, skin-to-skin contact, clean water, use of disinfectant, and good nutrition along with access to well-trained healthcare providers [9–11].

Many strategies were tried to halt these problems, which had an impact on neonates quality of life nationally and globally. The Millennium Development Goals (MDGs) included it, and later the Sustainable Development Goals (SDGs) focused on the MDGs "unfinished agenda. SDG 3's target 2 calls for all nations to make an effort to lower neonatal mortality to at least 12 per 1,000 live births by 2030 [6].

In Ethiopia, the neonatal mortality rate was reduced from 39 deaths per 1000 live births in 2005–29 deaths per 1000 live births in 2016, but not progressed since 2016 [5] and [12]. The 2019, Ethiopian Mini Demographic and Health Survey stated that the neonatal mortality rate was 30 deaths per 1,000 live births [5].

The national plan in this study area nation was to further reduce the neonatal mortality rate from 29 deaths per 1000 live births in 2016—10 deaths per 1000 live births by 2019/2020. To this effect, plans and strategies like the integrated management of newborn and childhood Illness (IMNCI), Kangaroo mother care (KMC), and Health sector development plan (HSDP) have been formulated [13,14].

Despite the aforementioned strategies and measures, Ethiopia is currently among the nations with a high reported number of neonatal fatalities in Africa. Although the neonatal mortality rate (NMR) in Ethiopia dropped from 49 per 1000 live births in 2000—29 per 1000 live births in 2016, the decline was substantially slower than that of post-neonatal under-five mortality between 2000 and 2016. The majority (about 75%) of neonatal deaths occur during the first week of life, according to national and international studies. In Ethiopia, out of 29 deaths per 1000 live births, about 21 deaths per 1,000 live births occur during the early neonatal period [5] and [15].

Therefore conducting a representative and methodologically sound investigation is required to pinpoint the root causes of the challenges and gaps mentioned. This is especially important in this study area of public hospitals where several socio-demographic factors are suspected. As such, this study was conducted to determine the incidence, time to death (event), and predictors of early neonatal mortality among early newborns admitted to neonatal intensive care units of the study area, in 2023.

Research evidence on survival and early newborn mortality predictors is now of extreme significance because it may be regarded as one of the best quality indicators for healthcare as well as for the social and economic welfare of the population. Additionally, such studies are uncommon, particularly in the central region of the country, despite 75% of neonatal deaths during the first week of life [16] and [17].

To better understand the incidence, predictors, and gaps preventing nations from achieving the Sustainable Development Goals (SDGs), which include a plan to eliminate preventable newborn deaths in all countries and reduce neonatal mortality to as little as 12 per 1000 live births, the study looks into these issues. Therefore, from this outcome, stakeholders working on the SDGs can get crucial evidence [6].

Evidence from this study will also give program implementors and policymakers actual data for monitoring and evaluation operations. The outcome of this study is crucial for healthcare professionals, particularly those working in clinics for maternal and child health. Additionally, the results of this study may serve as a benchmark for researchers to come to the same area of interest.

## 2. Methods and materials

A multi-centered retrospective cohort study was conducted in NICUs of four selected Public Hospitals in the study area from January 1, 2022, to January 1, 2023. The data were collected from January 30 to April 30, 2023.

This study was performed under the Declaration of Helsinki Ethical Principles for Medical Research involving human subjects protocol. Ethical clearance was obtained from Addis Ababa University, college of health sciences institutional review board (with IRB number of-27///2022/2023) before the start of the study. An official support letter was written to each hospital and permission for data collection was sought from the responsible authorities. Informed consent from each NICU data/record keeper was obtained. They were assured that there would be no harm to their records, the records were replaced into their original place after filling out the checklist, and the collected data were made anonymous with results taken in aggregates before the start of data collection and they had been so agreed. The methodology in this study followed STROCSS 2019 -statements included for the Strengthening the Reporting of Cohort Study in Surgery [18]. Research registry at https://www.researchregistry.com/browse-the-registry#home/ registered this study with UIN: researchregistry10212.

### 2.1. Source and study population

The source population were all neonates aged less than seven days of life admitted to the NICU of the study hospitals, while those systematically selected neonates who were admitted to the NICU of the study hospitals from January 1, 2022, to January 1, 2023, and who fulfilled the inclusion criteria were the study population. The study hospitals were randomly selected and the study population were proportionally allocated for each study hospital depending on situational analysis of their NICU admission records.

### 2.2. Eligibility criteria

All less than seven-day-old neonates who were admitted to the corresponding neonatal intensive care unit (NICU) of each selected hospital and who fulfilled medical record criteria were included, while those with age more than seven days and who had missed variables or variables of interest were excluded.

### 2.3. Sample size determination and sampling technique

**2.3.1. Sample size determination.** The sample size was calculated using double population proportion for predictors. Meconium aspiration syndrome of neonatal mortality predictor showed the largest possible sample size. Using open-source Epi Info software, the sample size was determined under the following assumptions: significance level (1-alpha) 95%, power (1-beta) 80, and the ratio of unexposed to expose in the sample is (1:1), taking into account the key predictors of Meconium aspiration syndrome, HIV status of the mother, various forms of pregnancy death, and neonatal temperature upon admission [19–21] (Table 1).

**2.3.2. Sampling technique.** Four hospitals were selected from a total of 12 public hospitals in the capital city by a lottery method and early neonatal record charts were allocated proportionally by their total early neonatal admission records to their neonatal intensive care units (NICU). Each study participant was selected by a systematic random sampling technique using skip intervals (K). A situational analysis and proportional allocation show that the sampling interval (K) was 20 (twenty) and the first study participant (random start) chart was selected using the lottery method from the respective NICU data records. Then, every other twenty neonatal charts were studied, until the required sample size was collected during the study period (Fig 1).

**Table 1. Sample size calculation summery for predictors of neonatal morality at NICU of study hospitals during study period.**

| Predictors' | P | AHR | Sample size with 10% non-response | References |
|---|---|---|---|---|
| Meconium aspiration syndrome | 17 | 2.14 | 375 | [19] |
| Temperature at admission($^0$c) | 71.4 | 2.68 | 269 | [20] |
| Multiple pregnancy | 15 | 3.96 | 119 | [21] |

P = percent of unexposed with outcome, therefore, the final sample size was determined to be n = 375, which is the maximum sample size used for analysis.

## 2.4. Variables of the study

The time to death of neonatal mortality was our dependent variable, while socio-demographic status of a neonate and a mother like newborn sex, neonatal age at admission, and maternal age, neonatal factors like: gestational age, birth weight, Apgar score, multiple gestation, indication for neonatal NICU admission, congenital anomalies, and start of breastfeeding, maternal and health related factors like parity, antenatal care (ANC) follow-up, HIV status, mode of delivery, maternal problems and place of delivery were independent variables.

## 2.5. Operational definitions

**Time:** Period from starting of observation until the occurrence of outcome of observation (death or censored).
**Event:** Death of a neonate at a specific time (day) within the 7 days of life.
**Censored:** Study participants who did not experience events of interest during the follow-up period.
**Discharged:** with improvement or stayed with admission beyond 7th days of life.

## 2.6. Data collection protocols and procedures

The data were collected using structured checklist data collection tool. The data collectors received training and orientation on how to fill out the checklist data collection tool. The usual secondary data handling protocols of the study hospitals like appropriate handling and replacing each chart to its original place after data collection were maintained.

The data collectors recorded the time to death of neonates, socio-demographic variables of a neonate and a mother like newborn sex, neonatal age at admission, and maternal age, neonatal factors like gestational age, birth weight, Apgar score, multiple gestation, indication for neonatal NICU admission, congenital anomalies, and the start of breastfeeding, maternal and health-related factors like parity, ANC follow-up, HIV status, mode of delivery, maternal problems and place of delivery. Supervisors and a principal investigator regularly checked for data collection procedures and the data collection tools for completeness and clarity during the data collection period. At the time of data collection, medical records that were missing or incomplete for variables of interest were excluded. The data was collected as planned and there was no loss to follow-up or outliers during data collection and entry into the software.

## 2.7. Data quality assurance

Before data collection started, a pretest was done on 5% of the sample size (19 patient charts) in a public hospital other than the study sites. Training was given to data collectors and supervisors regarding ethical issues, general approaches, and strategies to minimize information bias. A principal investigator cross-checked each questionnaire for completeness, accuracy, and consistency.

## 2.8. Data processing and analysis

The collected data were coded, cleaned, edited, and entered into Epi Data version 3.1 and then exported to SPSS software version 26.0 for analysis. Descriptive statistics like percentage, mean and standard deviation were employed as appropriate.

 

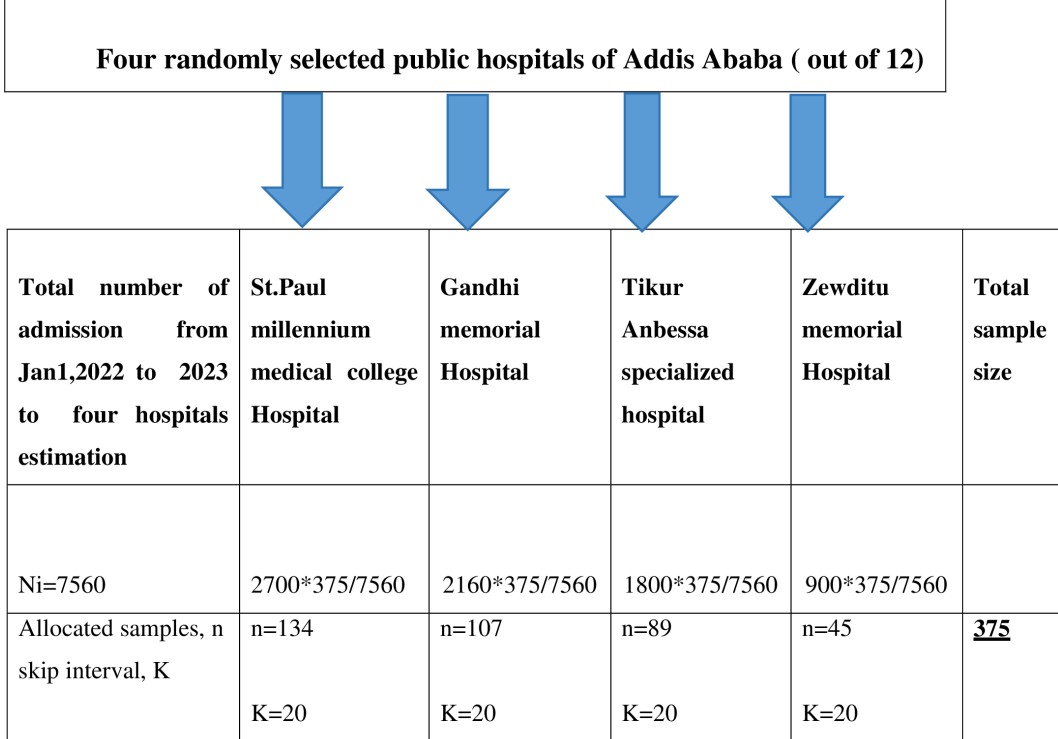

| Total number of admission from Jan1,2022 to 2023 to four hospitals estimation | St.Paul millennium medical college Hospital | Gandhi memorial Hospital | Tikur Anbessa specialized hospital | Zewditu memorial Hospital | Total sample size |
|---|---|---|---|---|---|
| Ni=7560 | 2700*375/7560 | 2160*375/7560 | 1800*375/7560 | 900*375/7560 | |
| Allocated samples, n skip interval, K | n=134 | n=107 | n=89 | n=45 | **375** |
| | K=20 | K=20 | K=20 | K=20 | |

Note that Ni is total early neonatal charts recorded in all the four NICU of the study hospitals during study period(from January 1. 2022 to January,1, 2023).

**Fig 1. Schematic presentation of sampling procedure for the study conducted in the NICU of study area during study period (n = 375).**

A necessary assumption of the Cox proportional hazard regression model was checked. Both bi variable and multivariable Cox proportional hazard regression analyses were computed. Moreover, the Kaplan-Meier survival curve was used to estimate survival time.

All variables having a P-value < 0.25 in the bi variable analysis were then further fitted to the final model to identify independent predictors of early neonatal mortality, and then the variables that had an independent association with an outcome variable were identified based on AHR with a 95% CI and a p-value of <0.05. As such, in the multivariable analysis, a P-value <0.05 was considered a predictor of early neonatal mortality. Frequencies, percentages, tables, and figures were used to present results.

## 3. Results

The data set for this study was extracted from the medical records of 375 study participants admitted to the NICU of study hospitals from Jan-1-2022 to Jan-1-2023.

### 3.1. Maternal and neonatal socio-demographic characteristics

A total of 375 neonatal charts admitted to the respective hospital NICUs were studied. The majority 199 (53.1%) were male. More than half of the participants, 281 (74.9%), were urban residents. Most neonates 362 (96.1%) were admitted to the NICU within the first three days or 72 hours of age. The majority of mothers, 279 (74.4%), were aged between 21–35 years. The minimum birth weight was 900 grams, while the maximum was 4500 grams. About 131 (35%) were born with a birth weight of less than 2500 grams (Table 2).

## 3.2. Maternal obstetric characteristics

Of the total 328 (87.5%) of mothers were multiparous, and 24 (6.4%) of the mothers were primiparous. About 326 (86.9%) mothers had ANC follow-ups, while 49 (13.1%) mothers had no prior history of ANC visits. The majority of mothers 313 (83.5%) gave birth in a healthcare facility. Nearly half 179(46.7%) of the mothers had a spontaneous vaginal delivery, while one-third 141 (37%) had a cesarean delivery. Three fourth 291 (77.6%) of newborns delivered were singleton. Pregnancy-induced hypertension was the most frequent obstetrical complication experienced during the previous pregnancy, followed by anemia during the current pregnancy 22 (5.9%), premature membrane rupture 35(9.3%), Ante-partum hemorrhage 12 (3.2%), HIV infection 62 (16.5%), diabetic mellitus 17 (4.5%), and others (8.5%) (Table 3).

## 3.3. Neonatal characteristics

Among the neonates admitted to the NICU of the study area; 158 (40.4%) and 80 (20.5%) of them had low first and fifth-minute Apgar scores respectively. Regarding the gestational age about 146 (38.9%) were preterm birth neonates. Concerning method of delivery, roughly half of the neonates, 179 (46.7%) had a spontaneous vaginal delivery (SVD); around one-third, 151 (37%) neonates were born through cesarean delivery. Throughout the admission stage to the newborn intensive care units, various reasons for neonatal admission were identified. Early neonatal sepsis 208 (55.5%), respiratory distress syndrome 111 (29.6%), perinatal asphyxia 169 (45.1%), neonatal jaundice 86 (22.9%), congenital anomalies 25 (9.4%), and preterm births 146 (38.9%) were among the common neonatal reasons for admission (Table 4).

## 3.4. Survival status of early neonatal life

At the end of the seventh day, of the total 375 participants, 75 (20%) were died 95% CI (15.7, 24.1). In addition, 293 (78%) neonates were released from hospitals, and 7(2%) were kept on therapy (Fig 2). A minimum of one day and a maximum of seven days with a median of 4 days and a mean of 3.4 days of follow-up time were recorded. Neonatal mortality is typically measured over 2025 person days, incidence rate of 37.1 deaths/1000 total person days was observed in this study.

The early neonatal survival rate during the first 24 hours, three days, and seven days of life was 95%, 86%, and 65% respectively. On the other hand, this study showed that 32 (42.7%), 40 (53%), and 3 (4%) of the early neonates were

Table 2. Socio-demographic characteristics of Neonates and their mothers at the study hospitals during study period (n = 375).

| Variables | Category | Frequency | Percentage |
|---|---|---|---|
| Sex of neonate | Male | 199 | 53.1 |
| | Female | 176 | 46.9 |
| Place of residence | Urban | 281 | 74.9 |
| | Rural | 94 | 25.1 |
| Neonatal age at admission | <24hr | 159 | 42.4 |
| | 1-3 days | 203 | 54.1 |
| | More than 3 days | 13 | 3.5 |
| Weight of neonates in grams | 1000 | 7 | 1.9 |
| | 1000-1500 | 16 | 4.3 |
| | 1500-2500 | 108 | 28.8 |
| | 2500-4000 | 236 | 62.9 |
| | >4000 | 8 | 2.1 |
| Maternal age | ≤20 | 32 | 8.5 |
| | 21-35 | 279 | 74.4 |
| | >35 | 64 | 17.1 |

**Table 3. Maternal and child health service related characteristics of mothers of neonates admitted to the NICU of study areas during study period (n = 375).**

| Variables | Category | Frequencies | Percentage |
|---|---|---|---|
| ANC follow up | Yes | 326 | 86.9 |
| | No | 49 | 13.1 |
| Numbers of visited | No visit | 47 | 12.5 |
| | One visit | 28 | 8.5 |
| | Two visit | 78 | 23.8 |
| | Three visit | 111 | 33.8 |
| | Four and more visit | 111 | 33.8 |
| Parity | Primiparous | 24 | 6.4 |
| | Multiparous | 328 | 87.5 |
| | Grand-multiparous | 23 | 6.1 |
| Place of delivery | Health institution | 313 | 83.5 |
| | Home | 62 | 16.5 |
| Type of pregnancy | Single | 291 | 77.6 |
| | Multiple | 84 | 22.4 |
| Mode of delivery | SVD | 179 | 47.7 |
| | instrumental delivery | 55 | 14.7 |
| | Cesarean section | 141 | 37.6 |
| HIV status | Positive | 62 | 16.5 |
| | Negative | 313 | 87.7 |
| PIH | Yes | 46 | 12.3 |
| | No | 329 | 87.7 |
| Other maternal complication | No other complications | 267 | 71.2 |
| | CARDIAC | 15 | 4.0 |
| | Diabetes | 17 | 4.5 |
| | ASTHMA | 7 | 1.9 |
| | PROM | 35 | 9.3 |
| | ANEMIA | 22 | 5.9 |
| | APH | 12 | 3.2 |
| Resuscitation status | Yes | 361 | 96.3 |
| | No | 14 | 3.7 |

PIH = pregnancy induced hypertension, PROM= Premature rupture of membrane, APH= Antepartum haemorrhage

died within the first 24 hours, between the first and third day, and after the third day of follow-up respectively. This finding shows that 72 (96%) died within the first 72 hours (or three days) of follow-up (Figs. 3–6, Table 5).

### 3.5. Predictors of early neonatal mortality

The bivariate analysis showed that gestational age, ANC visit, pre-eclampsia, mothers HIV status, type of birth, 1st minute APGAR score, 5th minute APGAR score, birth weight at admission, the start of exclusive breastfeeding, neonatal respiratory distress syndrome(RDS), presence of birth asphyxia, and parity were the factors associated with time to death(event) at a p-value of 0.25. The five-minute Apgar score, gestational age, pre-eclampsia, the start of exclusive breastfeeding, and low birth weight at admission were observed as independent predictors of newborn death(event) in the multivariable Cox proportional hazard model (Table 6).

**Table 4. Reasons for neonatal admission to the NICU of study hospitals during the study period (n = 375).**

| Neonatal character | Category | Frequency | Percentage |
|---|---|---|---|
| Gestational age | Term | 229 | 61.1 |
| | Preterm | 146 | 38.9 |
| Mode of delivery | SVD | 179 | 47.7 |
| | instrumental delivery | 55 | 14.7 |
| | Cesarean section | 141 | 37.6 |
| Breastfeeding was started | within one hour | 49 | 13.10 |
| | after one hour | 326 | 86.90 |
| Feeding of the neonate | only breast milk | 21 | 6.1 |
| | breast with addition | 125 | 33.3 |
| | Formula | 227 | 60.5 |
| Respiratory distress syndrome | Yes | 111 | 29.6 |
| | No | 264 | 70.4 |
| Did the neonate have a sepsis? | Yes | 208 | 55.5 |
| | No | 167 | 44.5 |
| Presence of prenatal asphyxia | Yes | 169 | 45.1 |
| | No | 206 | 54.9 |
| Presence of jaundice | Yes | 86 | 22.9 |
| | No | 289 | 77.1 |
| Congenital anomaly | Yes | 19 | 5.1 |
| | No | 356 | 94.9 |
| Specify types of anomaly | No other complication | 349 | 93.1 |
| | Cardiac | 16 | 4.3 |
| | Other | 10 | 2.6 |

The result of this study showed that premature neonates had a four times greater risk of dying than term [AHR: 4, 95% CI (1.234, 5.80)]. Newborns from pre-eclampsia mother had a 2.41-times higher risk of dying than those born to non pre-eclampsia mothers [AHR: 2.41, 95% CI (1.163, 4.980)]. Neonates with low fifth-minute Apgar scores had about 4 times higher chance of dying than those with normal Apgar scores [AHR: 3.93, 95% CI (1.76, 8.771)]. The risk of mortality was 3.7 times [AHR: 3.69, 95% CI (1.135, 12.015)] greater in those who did not start exclusive breastfeeding than in those who did, and the risk of death in those with low birth weight was 2 times [AHR: 2.01, 95% CI (2.10 (1.28, 3.43) higher than those with normal birth weight.

### 3.6. Assessment of model adequacy

The overall global test of the full Cox model was checked for proportional hazard assumption and it was met (p-value = 0.313). All covariate met the proportional-hazard assumption.

## 4. Discussion

This study determined the rate of early neonatal mortality and its predictors among neonates admitted to the NICUs of the study area hospitals.

The current study showed that throughout the study period, 75(20%) of the newborns admitted to the NICUs passed away. In the study hospitals of this study, the incidence density rate was 37.1 per 1000 neonate days of observation (95% CI 25.5, 40.9). This is greater than the study done in the general and comprehensive specialty hospital in Mekelle, which found that there were 22.45 deaths per 1000 newborn-days of monitoring [22]. The sample size, study methodology, or

■ discharge   ■ death   ■ on treatmement

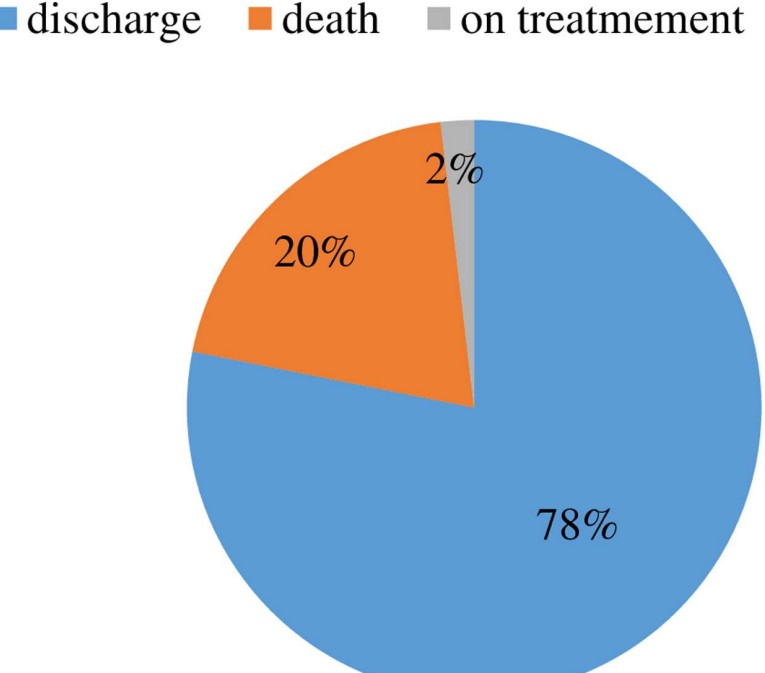

**Fig 2. Survival status of early neonatal life admitted to NICU of study area during study period (n = 375).**

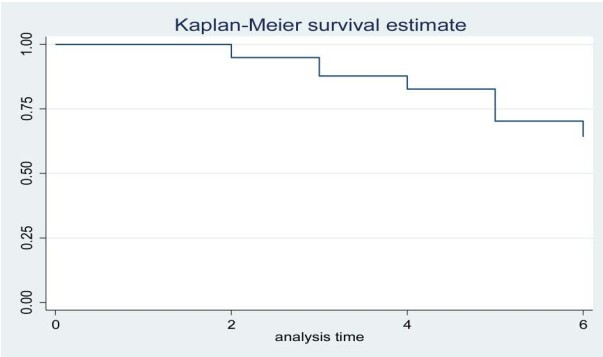

**Fig 3. Cumulative survival function analysis of Kaplan Meier for early neonatal life admitted to NICU of study area during study period (n = 375).**

population living standards could all be contributing factors to the difference. Additionally, hospitals outside this study area may refer more critical cases to this study area hospitals, which may increase the incidence rate of the event in the current study's referral hospitals. However, the finding of this study was consistent with the study done in Amhara regional state referral hospitals which reported early neonatal deaths at a rate of 34 per 1000 -days of observation, and Debre Markos referral hospital which reported early neonatal death at a rate of 39.6 per 1000 -days [23] and [24].

On the other hand, a study conducted in Wolayita Sodo observed a higher incidence rate of 77 new-born fatalities per 1000 neonate days [19]. The variation could be attributed to the populations living standards. In addition, the sample size, or study methodology could also be the possible contributing factors to the difference.

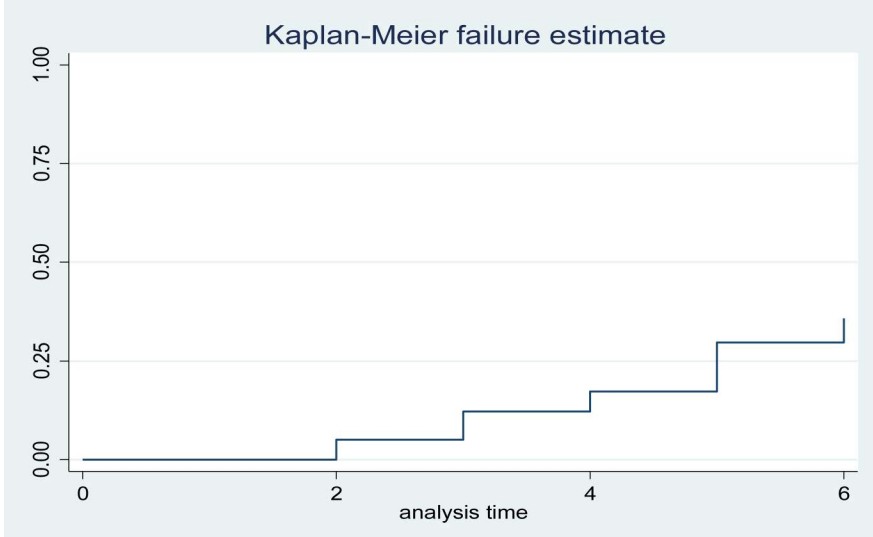

**Fig 4. Cumulative failure function analysis of Kaplan Meier for early neonatal life admitted to NICU of study area during study period (n = 375).**

The result of the current study showed that among those neonates, about 42.7% died within the first 24 hours, and about 53% of them died during the first and third days. This means that about 96% of neonatal deaths occurred within the first 72 hours (three days) of the follow-up. This finding is higher than the study conducted in Ghana which reported the highest death rate to be 76% within the first three days of follow-up [25]. However, this study showed a lower mortality rate than a study done in Amhara regional state referral hospitals which showed about 56.14% deaths within 24 hours of the predictors of neonatal death.

Additionally, this study showed a lower mortality rate than a study done in Gamo and Gofa public hospitals in southern Ethiopian regions which found that 46% of neonatal deaths happened within the first 24 hours and 44% happened between the first and third days [26]. However, the Tigray regional state study reported fewer fatalities than this amount, with around half (56%) of the fatalities occurring within the first three days of follow-up [7]. This discrepancy may be attributed to the fact that admissions to the current study hospitals were referred from other facilities and that they often arrive later than naturally born neonates. The other possible explanation may be that of initial treatment given to neonates after their admission to the NICU likely keeps them alive for longer than the first 24 hours of their life.

Neonatal death is reported to be more likely with low Apgar scores in the 1st and 5th minutes following birth. This study showed that neonates with low fifth-minute Apgar scores had about 4 times higher risk of dying than those who had normal Apgar scores. This is in line with the recent studies that showed low Apgar scores have a stronger prognostic value of neonatal mortality [20,27,28]. The connection between the two is justified by the fact that obstetric problems lead to poor Apgar scores. The newborn may exhibit symptoms like a slow heartbeat or no heartbeat, weak respiration or no breathing, and little flexion or no flexion if the baby has a low Apgar score, which is defined as seven or lower score out of ten.

This study revealed that neonates who did not begin exclusive breastfeeding had a higher early neonatal mortality rate of about 4 times than those who did. This result is coherent with findings from other investigations carried out at various locations and periods [29] and [30]. This can be justified as immediate exclusive breastfeeding serves as the starting point for ongoing mother and infant care, which can have long-lasting impacts on development and health.

Early breastfeeding prevents neonates from hypothermia, bacterial infection, and hypoglycemia, it also offers necessary nutrition at the proper time as well as immunological value from the first milk (colostrum). Neonatal death was shown to be lowered by 16% if breastfeeding was started within the first day of birth and by 22% if it was started within an hour

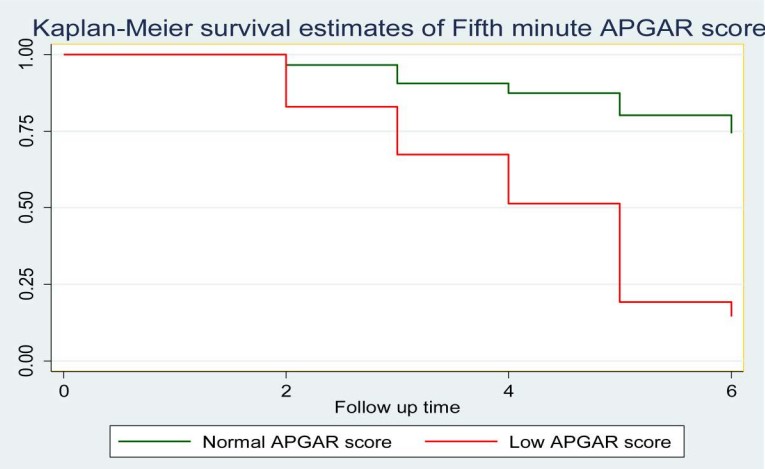

**Fig 5. Kaplan Meier survival estimate of the Apgar score for early neonates admitted to the NICU of study area during the study period (n =375).**

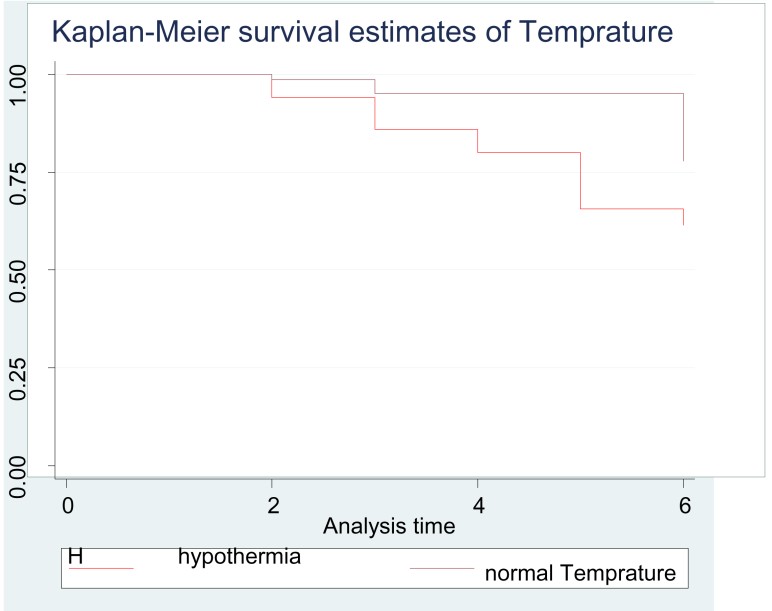

**Fig 6. Kaplan Meier survival estimate of the temperature of early neonates admitted to the NICU of study area during study period (n =375).**

after birth [31]. In addition, unwell neonates may not be able to suck breast milk like the neonates who are in better health condition at birth.

The survival risk of neonates born prematurely affects all nations globally, unlike the other factors that make children's lives [32] and [33]. The result of this study was consistent with that finding, which showed that the hazard of death for preterm babies was 3 times higher than for those born on the deadline (at term). This is also in line with findings from other investigations [34]. This may be because preterm infants are more likely to die due to physiologic and anatomical

**Table 5. Comparison of death among the predictor's variables using log-rank test for neonates admitted to the NICU of study hospitals during the study period (n = 375).**

| Variable | Log-rank (χ2) | p-value |
|---|---|---|
| Gestational age | 70. | <0.001 |
| Pre-eclampsia | 25.69 | <0.001 |
| 5th, APGAR score | 49.70 | <0.001 |
| Initiate EBF | 62.79 | <0.001 |
| 1st min. APGAR | 109.74 | <0.001 |

EBF = Early breastfeeding.

immaturity, which increases their chance of dying. Health issues caused by the underdevelopment of organs, muscles, and immune systems increase the risk for infants born preterm (before 37 weeks). Hence, they were more likely to experience difficulties like birth asphyxia, infections, and hypothermia, which leads to tissue hypoxia and multiple organ failure [35] and [36].

**Table 6. Predictors of time to death among neonates admitted to the NICU of study area during study period (n = 375).**

| Variables | Category | Survival status | | CHR | AHR |
|---|---|---|---|---|---|
| | | (Death %) | (Censored %) | | |
| Sex of neonate | Male | 34(17.1) | 165(82.9) | 0.65 (0.43, 1.03) | 0.68 (0.43,1.09) |
| | Female | 41(23.3) | 135(76.7) | 1 | 1 |
| Weight of neonates | 2500g or more | 36(14.8) | 208(85.2) | 1 | 1 |
| | Low birth weight | 39(29.8) | 92(70.2) | 2.26 (1.34, 3.36) | 2.10 (1.28,3.43) |
| Gestational age | term | 49(17.8) | 226(82.2) | 1 | 1 |
| | preterm | 26(26.0) | 74(74.0) | 7.02 (4.1,12.04 | 4 (1.234, 5.80) |
| HIV status | Positive | 16 (25.8) | 46(74.2) | 2.77 (1.02, 3.09) | 1.83(0.75, 3.20) |
| | Negative | 59 (18.8) | 254(81.2) | 1 | 1 |
| PIH | Yes | 14(30.4) | 32(69.6) | 3.34 (2.02,5.52) | 2.41 (1.16,4.98) |
| | No | 61(18.5) | 268(81.5) | 1 | 1 |
| Type of birth | multiple | 23(27.4) | 61(72.6) | 0.65(0.398,1.063) | 1.15 (0.59,2.56) |
| | single | 52(17.9) | 239(82.1) | 1 | 1 |
| Parity | Primiparous | 10(16.7) | 20(83.3) | 1.45 (0.9,2.35) | 1.47 (0.71, 3.07) |
| | multiparous | 65(19.8) | 263(80.2) | 1 | 1 |
| 1st min. AGPAR | >=7 | 21(6) | 196(94) | 1 | 1 |
| | <7 | 54(34.2) | 104(65.8) | 6.17 (3.43,11.10) | 1.10 (0.48,2.57) |
| 5th min.APGAR | >=7 | 25(8) | 270(92) | 1 | 1 |
| | <7 | 50(53.8) | 30 (46.3) | 8.72(5.32,14.3) | 3.93 (1.5,6.77) |
| Initiate EBF | Yes | 14(6.4) | 188(93.6) | 1 | 1 |
| | No | 61(31.2) | 112(68.8) | 5.2 (2.94,9.52) | 3.69 (1.14,9.45) |
| ARDS | Yes | 30(27.0) | 81(73.0) | 2.34(1.42,3.85) | 1.16 (0.6,1.75) |
| | No | 45(17.0) | 219(83.0) | 1 | 1 |
| Birth asphyxia | yes | 41(24.3) | 128(75.7) | 2.09 (1.25,3.50 | 1.1 (0.56,2.15) |
| | **No** | 34(16.5) | 172(83.5) | 1 | 1 |

Note: ARDS = acute respiratory distress syndrome, EBF = exclusive breastfeeding.

Note that results are from bi-variate and multivariate analysis using the Cox regression model for predictors, the significance level was taken at a p-value <0.05.

Pregnancy-induced hypertension (PIH) was another factor in the current study that predicted early neonatal mortality. Neonates who survived from pre-eclampsia mothers suffered a death risk that was 2.4 times greater than those who were born from non-pregnancy induced hypertension mothers. This result is in line with findings from other investigators [37] and [38]. Pregnancy-induced hypertension has been linked to an increased risk of low birth weight, small for gestational age (SGA) newborn, neonatal mortality, preterm, and other neonatal problems, according to the study.

Contrary to the other studies, ante-natal care (ANC) visits [19,26,39]. Multiple pregnancies [21,40,41], and place of residence [8] and [42] did not significantly predict neonatal death in the current study area. This could be due to the highest ANC coverage in the study area or due to socio-demographic differences between the study area and previous study sites [15,43,44].

## 4.1. Strengths and limitations of the study

The strength of this study is that the data collectors were well-trained professional nurses to ensure the accuracy of the collected data. The study was conducted in multi-center study hospitals. This study's consideration of censored observations contributed to another significant strength by giving a more precise estimate for the survival analysis. The predictor variables, which were recorded at admission, and the outcome covariate (death), were simple to sequentially associate. Since secondary sources were used to perform the study, some crucial maternal factors like birth interval, history of abortion, history of stillbirth, educational level, employment status, nutritional status, marital status, and economic status were not included as these factors may be important predictors of newborn death.

## 4.2. Implications and relevance

This research output can be an asset to improve pediatric patients healthcare quality by identifying the independent predictors of early neonatal death after NICU admission. It may also help as evidence for time to event (death) for early neonatal death in the NICU of the study region, where we have limited evidence so far. It offers researchers important insight, particularly for a future study.

## 4.3. Conclusion and recommendation

In this study area, the 5th minute Apgar score, preterm birth, low birth weight, inability for early initiation of exclusive breastfeeding, and pregnancy-induced hypertension were factors that predicted early neonatal time to death.

We recommend health care providers, mothers, families, and communities give due attention and quality health care to pregnant women with high blood pressure, newborns with low Apgar scores, and low birth weights, and babies who are not starting to breastfeed right away, as well as preterm neonates. We also recommend conducting further prospective longitudinal studies to address potential issues that could obstruct neonatal survival in the study area.

## Supporting information

**S1 Other. STROCSS guideline for strengthening the report of cohort studies in surgery 2019.**
(DOCX)

## Acknowledgments

We would like to express our gratitude to Addis Ababa University and each study hospital for granting ethical clearance and for allowing us to collect data respectively. We are also grateful for the data collectors, and record keepers who helped a lot with this research work.

## Author contributions

**Conceptualization:** Ashenafi Seifu Gesso, Gemechis Kabe Gonfa, Meron Abrar Awol.

**Data curation:** Ashenafi Seifu Gesso, Gemechis Kabe Gonfa, Meron Abrar Awol.

**Formal analysis:** Ashenafi Seifu Gesso, Gemechis Kabe Gonfa, Meron Abrar Awol.

**Investigation:** Ashenafi Seifu Gesso, Gemechis Kabe Gonfa, Meron Abrar Awol.

**Methodology:** Ashenafi Seifu Gesso, Gemechis Kabe Gonfa, Meron Abrar Awol.

**Project administration:** Ashenafi Seifu Gesso, Gemechis Kabe Gonfa, Meron Abrar Awol.

**Resources:** Ashenafi Seifu Gesso, Gemechis Kabe Gonfa, Meron Abrar Awol.

**Software:** Ashenafi Seifu Gesso, Gemechis Kabe Gonfa, Meron Abrar Awol.

**Supervision:** Ashenafi Seifu Gesso, Gemechis Kabe Gonfa, Meron Abrar Awol.

**Validation:** Ashenafi Seifu Gesso, Gemechis Kabe Gonfa, Meron Abrar Awol.

**Writing – original draft:** Ashenafi Seifu Gesso, Gemechis Kabe Gonfa, Meron Abrar Awol.

**Writing – review & editing:** Ashenafi Seifu Gesso, Gemechis Kabe Gonfa, Meron Abrar Awol.

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
