## [Editor Report · Decision Letter 0]

15 Aug 2024

PONE-D-24-21573SURVIVAL STATUS AND PREDICTORS OF EARLY NEONATAL MORTALITY AMONG NEONATES ADMITTED TO NEONATAL INTENSIVE CARE UNITS IN ADDIS ABABA PUBLIC HOSPITALS, ETHIOPIA 2023: A RETROSPECTIVE COHORT STUDY PLOS ONE

Dear Dr. Gesso,

Thank you for submitting your manuscript to PLOS ONE. After careful consideration, we feel that it has merit but does not fully meet PLOS ONE’s publication criteria as it currently stands. Therefore, we invite you to submit a revised version of the manuscript that addresses the points raised issues.

A rebuttal letter that responds to each point raised by the academic editor A marked-up copy of your manuscript that highlights changes made to the original version. You should upload this as a separate file labeled 'Revised Manuscript with Track Changes'.An unmarked version of your revised paper without tracked changes. You should upload this as a separate file labeled 'Manuscript'.

We look forward to receiving your revised manuscript.

Kind regards,

Mohammed Abdurke Kure, MSc, Assistant Professor

Academic Editor

PLOS ONE

Journal Requirements:

2. We note that your Data Availability Statement is currently as follows: "All relevant data are within the manuscript and its Supporting Information files"

Additional Editor Comments:

Dear Dr. Gesso,

Thank you for submitting the manuscript entitled "SURVIVAL STATUS AND PREDICTORS OF EARLY NEONATAL MORTALITY AMONG NEONATES ADMITTED TO NEONATAL INTENSIVE CARE UNITS IN ADDIS ABABA PUBLIC HOSPITALS, ETHIOPIA 2023: A RETROSPECTIVE COHORT STUDY" as Original Research Article. Your manuscript is suitable for the PLOS ONE Journal, for publication. However, it doesn't fulfill the publication criteria in its current stands. PLOS ONE is only considere the manuscript submitted with less typographic errors and standard English language. Please, revise the PLOS ONE authors'guidelines and re-submit your mansucript. For instance; your manuscript:

-Has no line-numbers

-Not double spaced

-You did not follow the PLOS ONE guidelines to prepare the Figures and Tables

-Has many typographic and editorial errors

Therefore, we can proceed with the manuscript, only after all these issues are resvised and resolved.

Regards

Mr. Mohammed Abdurke Kure, MSc, Assistant Professor

Academic Editor

PLOS ONE Journal

---

## [Author Response · Author response to Decision Letter 1]

29 Sep 2024

Dear, academic editors of the PLOSE ONE journal, This is Ashenafi Seifu Gesso a corresponding author for the manuscript entitled “Survival status and predictors of early neonatal mortality among neonatal intensive care unit admitted neonates in Addis Ababa, Ethiopia 2023: A retrospective cohort study”. It’s a great honor to work with this journal and I have publications previously as well. This journal is one of the most respected and reputable journals we are considering to publish our scholarly articles. As such, I am recommending friends, and colleagues to join working with PLOSE ONE journal. Below points are point by point responses for editors comments.

Response- I was not strictly followed Plose one authors guidelines at the prior submission. Now, in the revised submission I did it as per the journal standards.

2.We note that your Data Availability Statement is currently as follows: "All relevant data are within the manuscript and its Supporting Information files"

Response= In the revised version I stated it as “The datasets used and/or analyzed during the current study are available from the corresponding author without restriction”.

3.Your ethics statement should only appear in the Methods section of your manuscript.

Response=Yes, it was stated almost twice in the prior submission, but corrected in the revised submission.

4. Please include captions for your Supporting Information files at the end of your manuscript, and update any in-text citations to match accordingly

Response= The prior submission provided supporting information separately. The revised submission added captions for supporting information at the end of the manuscript file.

Additional comments

Please, revise the PLOS ONE authors' guidelines and re-submit your manuscript. For instance; your manuscript:

-Has no line-numbers

-Not double spaced

-You did not follow the PLOS ONE guidelines to prepare the Figures and Tables

-Has many typographic and editorial errors

Response

The former submission had no line-numbers, revised one has continuous line numbers

The prior submission was 1.5 line spaced, but the revised one is double spaced

The former submission did not followed captions for tables and figures, but the revised one was inline with PLOS ONE guidelines to prepare the Figures and Tables

The first submission had some grammatical and punctuation errors, but the revised one solved those typographic and editorial errors.

---

## [Decision Letter · Decision Letter 1]

2 Mar 2025

PONE-D-24-21573R1SURVIVAL STATUS AND PREDICTORS OF EARLY NEONATAL MORTALITY AMONG NEONATAL INTENSIVE CARE UNIT ADMITTED NEONATES IN ADDIS ABABA, ETHIOPIA 2023: A RETROSPECTIVE COHORT STUDYPLOS ONE

Dear Dr. Gesso,

Thank you for submitting your manuscript to PLOS ONE. After careful consideration, we feel that it has merit but does not fully meet PLOS ONE’s publication criteria as it currently stands. Therefore, we invite you to submit a revised version of the manuscript that addresses the points raised during the review process by the reviewers comments and suggestions. 

 Please submit your revised manuscript by Apr 16 2025 11:59PM. If you will need more time than this to complete your revisions, please reply to this message or contact the journal office at plosone@plos.org . Please include the following items when submitting your revised manuscript:

We look forward to receiving your revised manuscript.

Kind regards,

Dawit Getachew 

Academic Editor

PLOS ONE

Reviewers' comments:

Reviewer's Responses to Questions

**Comments to the Author**

1. If the authors have adequately addressed your comments raised in a previous round of review and you feel that this manuscript is now acceptable for publication, you may indicate that here to bypass the “Comments to the Author” section, enter your conflict of interest statement in the “Confidential to Editor” section, and submit your "Accept" recommendation.

Reviewer #1: All comments have been addressed

Reviewer #2: (No Response)

2. Is the manuscript technically sound, and do the data support the conclusions?

Reviewer #1: Yes

Reviewer #2: No

3. Has the statistical analysis been performed appropriately and rigorously? 

Reviewer #1: Yes

Reviewer #2: No

4. Have the authors made all data underlying the findings in their manuscript fully available?

Reviewer #1: Yes

Reviewer #2: No

5. Is the manuscript presented in an intelligible fashion and written in standard English?

Reviewer #1: Yes

Reviewer #2: No

6. Review Comments to the Author

Reviewer #1: All my comments and suggestion given in the first version, addressed in this version, no adtional comments, concerns and suggestion on this version.

Reviewer #2: Dear editor I would like to thank you for inviting me to review this paper done by Gesso et. al. The paper tried to investigate incidence and predictors of mortality of neonates in neonatal ICU in Addis Ababa, Ethiopia, using a retrospective chart review. This manuscript brings unaddressed challenge of public health. However, the manuscript lacks fundamental point to merit publication.

Major issues

1. Lack of novelty: predictors of neonatal mortality in NICU have been well studied in various parts of Ethiopia including Addis Ababa. A study done by Getahun et. al (https://www.frontiersin.org/journals/pediatrics/articles/10.3389/fped.2024.1352270/full) assessed predictors of neonatal mortality in NICU using similar settings, study design, and analysis technique, making this manuscript a duplication of results.

2. Study design: Almost all previously conducted studies used retrospective cohort study (chart review) recommended to further conducting prospective studies in order to better estimate incidence and identify predictors of neonatal mortality in the NICU. However, this recent manuscript used retrospective chart review which makes the manuscript less attractive.

3. Length of follow-up in NICU: The authors described neonates stayed in NICU a minimum of 1 day to a maximum of 7 days. This seems unrealistic since neonates, especially on phototherapy, sepsis, and neonatal enterocolitis, might stay more than 7 days. This raises a question on data collection.

4. Result and Statistics: A poor presentation of results. It is not relevant to provide Log-rank test results id the authors run Cox regression. Not relevant to upload Kaplan-Meier graphs of categorical variables as a supplementary file. The authors did not assess variability across clinical settings using a statistical approach (This might lead to analysis with shared frailty model).

In conclusion lack of novelty makes this manuscript less attractive. I recommend the author to consider prospective study.

7. PLOS authors have the option to publish the peer review history of their article (what does this mean? ). If published, this will include your full peer review and any attached files.

**Do you want your identity to be public for this peer review?** For information about this choice, including consent withdrawal, please see our Privacy Policy .

Reviewer #1: **Yes: ** Esubalew Tesfahun

Reviewer #2: No

---

## [Author Response · Author response to Decision Letter 2]

6 Apr 2025

Point by point response to the editors and reviewers comments

1. If the authors have adequately addressed your comments raised in a previous round of review and you feel that this manuscript is now acceptable for publication.

Response:  Yes we appropriately revised as per the comments given in the subsequent reviews .

2. Is the manuscript technically sound, and do the data support the conclusions?

Response: Yes, absolutely the conclusions were drawn from the results presented.

3. Has the statistical analysis been performed appropriately and rigorously?

Response:  Yes, we performed survival analysis to see when will a neonate less than or equal to 7 days old would die after admission to neonatal intensive care units.

4. Have the authors made all data underlying the findings in their manuscript fully available?

Response: Yes, we declared it in detail and we can provide every data sets used for this analysis without restriction.

5. Is the manuscript presented in an intelligible fashion and written in standard English?

Response: Yes, we made this manuscript well articulated and informative through our publication experience and PLOS ONE standards.

6. Review Comments to the Author

Reviewer #1: All my comments and suggestion given in the first version, addressed in this version, no additional comments, concerns and suggestion on this version.

Reviewer #2: Dear editor I would like to thank you for inviting me to review this paper done by Gesso et. al. The paper tried to investigate incidence and predictors of mortality of neonates in neonatal ICU in Addis Ababa, Ethiopia, using a retrospective chart review. This manuscript brings unaddressed challenge of public health. However, the manuscript lacks fundamental point to merit publication.

Major issues

1. Lack of novelty:

Response:  We see the concerns over the novelty, but the previous study was simple cross sectional study with incidence and associated factors for NICU admitted neonates. The current study(ours) was survival study that looks when will the early neonate (from first day to day seven of life) die after admission to the NICU. Ou study was different in population, in methods, analysis and finding as well.

2. Study design: Almost all previously conducted studies used retrospective cohort study (chart review) recommended to further conducting prospective studies in order to better estimate incidence and identify predictors of neonatal mortality in the NICU. However, this recent manuscript used retrospective chart review which makes the manuscript less attractive.

Response: The current study (our was also retrospective), we understand and we stated the limitations of retrospective study in the methods section of the manuscript. But the objective was to pick the predictors of early neonatal mortality (≤7 days of life) not for all neonates admitted to the neonatal intensive care units.

3. Length of follow-up in NICU: The authors described neonates stayed in NICU a minimum of 1 day to a maximum of 7 days. This seems unrealistic since neonates, especially on phototherapy, sepsis, and neonatal enterocolitis, might stay more than 7 days. This raises a question on data collection.

Response: With regard to this concern, we put early neonatal mortality in operational definition in the manuscript. Accordingly, if a neonate stays for more than seven days we declared improved or stayed beyond 7th days of life, if not an event was recorded as died.

4. Result and Statistics: A poor presentation of results. It is not relevant to provide Log-rank test results if the authors run Cox regression. Not relevant to upload Kaplan-Meier graphs of categorical variables as a supplementary file. The authors did not assess variability across clinical settings using a statistical approach (This might lead to analysis with shared frailty model).

Response:We understand the concern from the reviewer, but we added Kaplan-Meier graphs to increase clarity for the readers and to see time to death as long as this would not alter the cox-regression findings.

---

## [Editor Report · Decision Letter 2]

11 Apr 2025

SURVIVAL STATUS AND PREDICTORS OF EARLY NEONATAL MORTALITY AMONG NEONATES ADMITTED TO NEONATAL INTENSIVE CARE UNITS IN ADDIS ABABA, ETHIOPIA 2023: A RETROSPECTIVE COHORT STUDY

PONE-D-24-21573R2

Dear Dr. Ashenafi,

We’re pleased to inform you that your manuscript has been judged scientifically suitable for publication and will be formally accepted for publication once it meets all outstanding technical requirements.

Kind regards,

Dawit Getachew Gebeyehu, MPH

Academic Editor

PLOS ONE
---

## [Editor Report · Acceptance letter]

PONE-D-24-21573R2

PLOS ONE

Dear Dr. Gesso,

I'm pleased to inform you that your manuscript has been deemed suitable for publication in PLOS ONE. Congratulations! Your manuscript is now being handed over to our production team.

Kind regards,

on behalf of

Mr. Dawit Getachew Gebeyehu

Academic Editor

PLOS ONE